# Peer review of "The Role of AI in Hospitals and Clinics: Transforming Healthcare in the 21st Century"

_bioengineering, 2024, doi:10.3390/bioengineering11040337_

Round 1

Reviewer 1 Report

Comments and Suggestions for Authors

The paper proposes a comprehensive survey of AI-driven solutions for healthcare. The paper is well structured and covers most of the relevant aspects of this field.

The paper could be improved in few directions.

1) The paper provides a large amount of information. The paper presentation should be improved by introducing an image depicting the structure of content of each core section of the paper. Moreover, some illustration depicting the physical tools used in AI-driven healthcare would make the read more enjoyable.

2) The authors should explain which methodology they used to produce the survey; e.g., which strategies have been used to search and select the most relevant papers and projects reported in the manuscript.

3) Acceptance of patients and clinicians is a key factor for the impact of these solution. This aspect is only marginally considered in the paper. A more thorough discussion of stakeholders' acceptance should be added to the manuscript.

4) A more detailed discussion of user-centered design and of methodologies to evaluate AI-based healthcare solutions should be provided.

Author Response

Thank you for taking the time to review our manuscript. We appreciate your valuable feedback and have made the following improvements in response to your comments:

  1. Inclusion of a Figure: To enhance the readability and organization of the paper, we have modified figure 1 to include the details covered in the paper. This modification provides readers with a clear overview of the paper's structure, making it easier to navigate through its core sections.

  2. Expansion on Hardware Acceleration: We have addressed the suggestion to include information about hardware accelerators by providing further details in section 4.3, "The Role of Hardware Acceleration in AI-powered Diagnostics," and section 8.2.

  3. Adding extra table to Section 5 for better clarity.

We believe that these enhancements will address the concerns raised and improve the overall quality of the manuscript.

2-We have now included details about our search methodology in the Introduction Section of the manuscript. It reads as follows:

"In consideration of the extensive scope of this study, we employed a meticulous approach in selecting references, focusing primarily on works published within reputable journals over the past five years. Our search was conducted using both Google Scholar and PubMed, ensuring a comprehensive exploration of pertinent literature."

3- We agree that the acceptance of AI-driven solutions by patients and clinicians is crucial for their successful implementation in healthcare settings. To address this concern, we have added a new section (8.5.2) specifically dedicated to discussing Stakeholder Acceptance. In this section, we explore the various factors influencing the acceptance of AI solutions among patients and clinicians, including trust, usability, and ethical considerations. 

4- We have incorporated a more detailed discussion of user-centered design methodologies into the manuscript under section 6.2. Interpretability and Usability. In this section, we explore how user-centered design principles can enhance the usability and effectiveness of AI-based healthcare solutions. 

Reviewer 2 Report

Comments and Suggestions for Authors

he documentation is huge, fluent, readable and well-organized. However, despite a broad introduction and differentiation of the many aspects of AI applications in healthcare and medicine in general, the Conclusion section is too short. The work can be better balanced from the introduction to the conclusion.

Secondly, despite a very broad description of the variety of tools on how AI is provided, nothing is spent on hardware accelerators. Today AI applications are also optimized and tuned for the use of hardware accelerators that make use of FPGAs and GPUs, not only to be simulated in software. Thus, I suggest at least mentioning, within an additional section, the potential and advantages of using AI software running on hardware accelerators instead of using CPUs. This speeds up machine/deep learning processes by orders of magnitude.

Also, there are many acronyms and some are not spelled as:

- line 109-110: spell out MRI and VGG as encountered for the first time

- line 118: spell out BERT and GPT as encountered for the first time

- line 176: T2D

Author Response

1- Thank you for your feedback and suggestions. We have taken your comments into consideration and made several improvements to enhance the structure and balance of the paper.

  1. Figure 1: We have modfied figure 1 to summarize the topics covered in the paper and to provide better clarity and navigation through the paper. This will help readers quickly locate specific sections of interest.

  2. New Subsections and Sections: To cover additional details and ensure comprehensive coverage of AI applications in healthcare, we have introduced new subsections and sections. For instance, we have included a dedicated section on "Methodologies for Assessing AI Healthcare Solutions" to discuss critical evaluation methods. We have also removed unnecessary or redundant components to streamline the paper's focus.

  3. Balanced Conclusion: We recognize the importance of a well-developed conclusion that summarizes key insights and provides a cohesive ending to the paper. In response to your feedback, we have ensured that critical future directives and concluding paragraphs are distributed throughout different sections of the paper. Additionally, we have extended the conclusion section to provide a more comprehensive summary of the findings and implications discussed in the paper.

  4. Dedicated Sections for Ethical Considerations and Future Outlook: To address ethical concerns and explore future directions, we have included dedicated sections titled "Ethical Considerations and Challenges" and "The Future of AI in Healthcare." These sections provide in-depth discussions on the ethical implications of AI adoption in healthcare and highlight potential future trends and advancements.

  5. New table in Section 5 for better clarity.

2- Thank you for highlighting the importance of hardware accelerators in optimizing AI applications, specifically in enhancing processing power and speeding up machine/deep learning processes. We have addressed your suggestion by incorporating additional details in two specific sections of the paper:

  1. Section 4.3: The Role of Hardware Acceleration in AI-powered Diagnostics: We have introduced a new subsection dedicated to discussing the potential and advantages of using hardware accelerators in AI-powered diagnostics. This section elaborates on how hardware accelerators, such as FPGAs and GPUs, can significantly improve processing speed and efficiency, thereby enhancing the performance of AI algorithms in diagnostic applications.

  2. Section 8.2: Enhanced Treatment Technologies: We have also included information about hardware accelerators within the context of future directions and advancements in AI healthcare solutions. By highlighting the importance of optimizing AI software for hardware accelerators, we emphasize the potential for further improving the scalability and performance of AI-powered healthcare systems.

3- We have fixed the acronyms.

Reviewer 3 Report

Comments and Suggestions for Authors

The paper presents AI's role in healthcare in a broad perspective, including hospital operations and management, medical imaging, and patient care and monitoring. I am not an expert in all these research areas. I have some experiences in research studies related to the use of deep learning for cardiac and brain MR image analysis. However, since I am recently involved in eduation and research for the field of AI healthcare, the paper really helped me to know recent AI research studies on other research fields with which I am not familiar. If there are other readers like me, this review paper would draw attention, I think.  I feel that categorizing and structuring in "Medical Imaging" part require major modifications. My specific comments are the following.

1. (page 5, indentation in line 149-164) I think it is better not to indent these lines. Instead, I suggest the formatting as follows.

2.2.1. Cancer Detection

    One of the most groundbreaking applications ....

2.2.2. Diabetes Management

    AI's role in managing and detecting ....

2. Table 3 should be revised, I think. First, 'Imaging Modality' is not an appropriate word because it contains "Automated Image Segmentation". If the authors want to use 'Imaging Modality' in the table, it is better to categorize it into 'MRI', 'CT', 'ultrasound', 'X-ray', etc.

3. (page 17, 4.3 Examples of AI Systems Used in Imaging) There are six examples, and they seem to be related to Table 3. I think the six categories need to be modified so that one category is independent of the other category. In fact, I am sure that "AI in Cardiovascular Imaging" can include "Automated Image Segmentation in Radiology" tasks.  

Author Response

Thank you for your feedback and sharing your perspective on the paper. We appreciate your input and are glad to hear that the review paper has been helpful in broadening your understanding of AI research in healthcare. We have taken note of your comments regarding the categorization and structure of the "Medical Imaging" section. In response, we have revised and restructured this section to ensure clarity and coherence, addressing your specific concerns. Please find our response to your comments: 

  1.  It seems that the formatting of the paper, including the indentation of subsubsection lines, has been standardized by the journal's editing process. Therefore, the indentation style you mentioned is consistent with the subsubsection formatting used throughout the paper.
  2. Table 2 has been revised to include image modalities including MRI, CT, X-Ray, and Ultrasound.
  3. The text has been edited to follow the table format.

Reviewer 4 Report

Comments and Suggestions for Authors

Thanks for your valuable review on the roles of AI in the health care system. It is well-written and presents an overview of the fields. Congratulations to authors for this panoramic view. However, while authors have done a wonderful job, I believe this manuscript benefits from some improvements. One aspect One aspect that could be addressed is the inclusion of a summary and risk assessment associated with AI in each section, which may not be necessary. Additionally, there are sentences that are repeated in multiple sections, which could be removed and condensed. As a suggestion, I recommend the authors to create a table under each section, where they can describe related scientific reports (in each row) accompanied by the authors' names and publication years.

Minor Comments:

1. Generally, in the introduction, figures are not mentioned. Yes, it was a comprehensive overview of your paper, which is needed to be explained. You can refer in the section 2, where you have mentioned "this section will explore three critical aspects". 

2. Line 93: AI can predict the potential risk, but not manage, which can be managed by health care professionals. Thus, correct your statement. 

3. Line 110: What is ResNet and VGG?

4.  Please add another advanced CNN method in addition to GAN: diffusion model and cite the following paper: "DiffGEPCI: 3D MRI Synthesis from mGRE Signals using 2.5D Diffusion Model".

5. Line168: What are the 889 records? medical records or EEG recordings?

6. Very fine summary of about the case studies of AI in detecting diagnosis. Authors are also touched up on the potential risks and challenges. However, authors did not mention about the prediction loss. In health care system, radiologists are responsible for their errors in diagnosis, while in the computerized diagnosis who will be responsible for the prediction errors.

6. For example the AI predict the AD in ADNI cohort (mostly western population), can it be useful tool in diagnosis in other datasets from other regions. 

7. Line 342: "They complement rather than replace human expertise". It is an excellent suggestion as we hear from the day of AI domination in diagnosis. Would you please elaborate how it can be complement when AI performs better diagnostics (in identifying false positive or false negative) when compared with radiologist?

8. Line 442-443: "no-show rate decreased from 19.3% to 15.9% over six months, 442 marking a 17.2% improvement". The difference in increased patient show up is: 3.4%, then where is 17.2% coming from?

9. Line: 537-538: Authors, suddenly introducing new technical words ("particularly when using bias-adjusted Kalman filter step-sizes") in the flow, which is difficult to follow for readers unless they are familiar.

10.  Line 649-658: I am not sure this example fit in this context. In this section, you are focusing on using images to reducing the diagnostic errors. 

11. Line 760: Automated segmentation in radiology, in what images they do segmentation. Do they segment the tissue on images other than MRI, CT, PET, X-ray and Ultrasound? This paragraph is not necessary, please remove it. Also same with pediatric imaging. 

12. The Figure 2 is not mentioned in the text. Please provide the details about purpose of Figure 2. 

13. Line 954: Please number it: "Integration with Existing Clinical Workflows"

Author Response

Thank you for your thoughtful feedback on our review of the roles of AI in the healthcare system. We appreciate your acknowledgment of the overview presented in the manuscript. While we strive for clarity and conciseness in our writing, we tried to have a critical summary at the end of each section. Additionally, we have dedicated a full section to Ethical Considerations and Challenges (Section 7), where we extensively discuss the potential risks associated with AI in the healthcare system. Furthermore, due to the length constraints of the paper, we opted to enhance readability by modifing Figure 1 to include the topics coveted in the paper. Additionally, we have taken steps to remove redundant text and organize the content more efficiently. A new table has been added to Section 5 for better clarity. We hope these revisions address your concerns and improve the overall quality of the manuscript. Thank you again for your valuable input. Please find a detailed response to your comments: 

1- While we understand your suggestion to reference figures in Section 2, where specific aspects are explored, we believe that for a comprehensive overview of a lengthy review paper, it is necessary to provide clarity on the covered topics in the introduction. Therefore, we found the introduction to be the most suitable place to include this figure. However, we ensured that in Section 2 and throughout the paper, we provide clear and detailed explanations of the critical aspects mentioned, as suggested.

2- The statement has been corrected accordingly.

3-  In line 110, "ResNet" stands for "Residual Neural Network" and "VGG" refers to the "Visual Geometry Group" network. We have ensured to use the full names to provide clarity for readers who may not be familiar with these acronyms.

4- We have added another advanced CNN method, the diffusion model, in addition to GAN. Additionally, we have cited the paper titled "DiffGEPCI: 3D MRI Synthesis from mGRE Signals using 2.5D Diffusion Model" as per your recommendation.

5- We have further clarified this in the revised manuscript as follow: "The dataset utilized in this study comprises 79 input attributes, including results of medical tests and demographic information collected from 884 patients."

6- We have addressed this under section 7.1.4:

Transparency and Accountability: Transparency in AI decision-making processes is a key ethical concern. It is important for healthcare providers and patients to under-stand how AI systems make their recommendations. This transparency is essential for building trust in AI systems and for accountability. In cases where AI-driven decisions impact patient care, it is crucial to have mechanisms in place to review and understand these decisions, particularly in the event of adverse outcomes. A recent study highlights the need for transparent and accountable AI systems in natural NLP to address the "black box" issue of deep learning models. It introduces the Explaining and Visualizing CNNs for Text Information (EVCT) framework, which offers human-interpretable solutions for text classification with minimal information loss, aligning with recent demands for fairness and transparency in AI-driven decision support systems

Regarding the second comment about ADNI, we have addressed this under section:

6.2.

Interpretability and Usability

To earn trust and acceptance within the healthcare system, AI technologies must be interpretable, usable, and ethically sound. Interpretability ensures AI models provide clear explanations for their decisions, fostering trust with clinicians who can understand the reasoning behind recommendations 153. Usability focuses on seamless integration of AI tools into existing workflows for all stakeholders. User-centered design principles, with active involvement from clinicians and patients throughout development, are crucial not only for usability but also for user engagement. This collaborative approach fosters a sense of ownership and trust in the AI solution, ultimately driving successful adoption and improved patient outcomes.

Furthermore, interpretability extends beyond simply understanding the "why" behind an AI decision. Explainability techniques like feature importance analysis, LIME (Local Interpretable Model-agnostic Explanations)154, and SHAP (SHapley Additive exPlanations)155 values can provide deeper insights into the model's reasoning.

While interpretability and usability are crucial for initial acceptance of AI solutions, user engagement plays a vital role in driving long-term trust and successful adoption 156. User engagement refers to the ongoing interaction and positive user experience with the AI tool. Here's how user-centered design principles promote engagement:

  1. Active stakeholder involvement: Throughout the development process, actively involving clinicians, patients, and other stakeholders provides valuable insights into their needs and expectations. This collaborative approach fosters a sense of ownership in the solution, leading to higher engagement.
  2. Iterative development and feedback loops: Developing AI solutions is an iterative process. By incorporating user feedback throughout development cycles, researchers can refine the AI tool to better address user needs. This ongoing feedback loop not only improves usability but also strengthens user confidence and engagement.
  3. User-friendly interfaces and clear visualizations: Designing clear and user-friendly interfaces is essential for user engagement. This includes presenting AI outputs in a way that is easy to understand and interpret, even for users with limited technical expertise. Additionally, providing clear visualizations of the AI's reasoning can further enhance user trust and engagement.

And

Section 8.4.2. Ensuring Model Versatility: Achieving versatility in AI models is essential for their effective application across the diverse landscape of healthcare settings and patient demographics. Techniques such as domain adaptation and transfer learning stand out as effective solutions, enabling AI models trained on one dataset to adjust and perform accurately on another with little need for retraining. This capability is par-ticularly valuable in healthcare, where patient characteristics, disease profiles, and treatment responses can vary widely. By fostering such adaptability, these tech-niques ensure that AI can be deployed more universally, enhancing its effectiveness and utility for a broad spectrum of patients.

7- Regarding the complementarity of AI and human expertise, it's important to recognize that while AI models may outperform human experts in certain aspects of diagnostics, they are not intended to replace human judgment entirely. Instead, they serve as valuable tools that complement and enhance the capabilities of healthcare professionals. For example, AI excels in tasks such as identifying patterns in medical imaging with high accuracy and efficiency, including detecting subtle anomalies that may be overlooked by human observers. However, human expertise remains indispensable in interpreting complex clinical scenarios, considering patient history, and making informed decisions based on contextual factors that AI may not capture. Additionally, AI's ability to provide quantitative assessments and assist in decision-making can augment the diagnostic process, empowering clinicians to make more informed treatment decisions. Therefore, the integration of AI into healthcare workflows should be viewed as a symbiotic relationship, where AI supports and augments human expertise, ultimately leading to improved patient outcomes.

Our text has been modified accordingly.

8- Corrected.

9- Revised.

10- The example has been removed.

11- These paragraphs have been removed and place at the correct sections.

12- The section is numbered in the revised manuscript.

Round 2

Reviewer 1 Report

Comments and Suggestions for Authors

The authors have addressed most of my concerns.